# Validation of the Water Vapor Profiles of the Raman Lidar at the Maïdo Observatory (Reunion Island) Calibrated with Global Navigation Satellite System Integrated Water Vapor

**Hélène Vérèmes [1,2,\*]**, **Guillaume Payen [2]**, **Philippe Keckhut [3]**, **Valentin Duflot [1,2]**, **Jean-Luc Baray [4]**, **Jean-Pierre Cammas [1,2]**, **Stéphanie Evan [1]**, **Françoise Posny [1]**, **Susanne Körner [5]** and **Pierre Bosser [6]**

[1] Laboratoire de l'Atmosphère et des Cyclones, UMR8105 (CNRS, Université de La Réunion, Météo-France), 97490 Saint-Denis de La Réunion, France; valentin.duflot@univ-reunion.fr (V.D.); jean-pierre.cammas@univ-reunion.fr (J.-P.C.); stephanie.evan@univ-reunion.fr (S.E.); francoise.posny@univ-reunion.fr (F.P.)

[2] Observatoire des Sciences de l'Univers de La Réunion, UMS3365, 97490 Saint-Denis de La Réunion, France; guillaume.payen@univ-reunion.fr

[3] Laboratoire ATmosphères, Milieux, Observations Spatiales-IPSL, UMR8190 CNRS, UVSQ, UPMC, 78280 Guyancourt, France; Philippe.Keckhut@latmos.ipsl.fr

[4] Laboratoire de Météorologie Physique, UMR6016, Observatoire de Physique du Globe de Clermont-Ferrand, Université Clermont Auvergne, 63178 Clermont-Ferrand, France; J.L.Baray@opgc.fr

[5] Deutscher Wetterdienst, Meteorological Observatory Lindenberg, 15848 Lindenberg, Germany; susanne.koerner@dwd.de

[6] ENSTA Bretagne – Lab-STICC UMR CNRS 6285—PRASYS Team, 29200 Brest, France; pierre.bosser@ensta-bretagne.fr

\* Correspondence: helene.veremes@univ-reunion.fr

**Abstract:** The Maïdo high-altitude observatory located in Reunion Island (21°S, 55.5°E) is equipped with the Lidar1200, an innovative Raman lidar designed to measure the water vapor mixing ratio in the troposphere and the lower stratosphere, to perform long-term survey and processes studies in the vicinity of the tropopause. The calibration methodology is based on a GNSS (Global Navigation Satellite System) IWV (Integrated Water Vapor) dataset. The lidar water vapor measurements from November 2013 to October 2015 have been calibrated according to this methodology and used to evaluate the performance of the lidar. The 2-year operation shows that the calibration uncertainty using the GNSS technique is in good agreement with the calibration derived using radiosondes. During the MORGANE (Maïdo ObservatoRy Gaz and Aerosols NDACC Experiment) campaign (Reunion Island, May 2015), CFH (Cryogenic Frost point Hygrometer) radiosonde and Raman lidar profiles are compared and show good agreement up to 22 km asl; no significant biases are detected and mean differences are smaller than 9% up to 22 km asl.

**Keywords:** Raman lidar; water vapor; calibration; GNSS IWV; Maïdo Observatory; Reunion Island

---

## 1. Introduction

Observations of essential climate variables (ECV), such as atmospheric water vapor [1], are necessary to monitor potential climate changes. The factors influencing the spatiotemporal variability of this greenhouse gas are various—evaporation [2], convection [3–5], precipitation [6], temperature [7], transport [8,9] and dynamical processes from eddies to synoptic scale events [2,10]. The residence time

of water vapor in the atmosphere is several days to weeks [11,12]. Nevertheless, in the troposphere, the temporal and spatial variabilities of the water vapor can be high at a scale of dozens of minutes and less than one kilometer [2,13] and are considerably higher in the upper troposphere (UT) than in the lower stratosphere (LS). Water vapor is a challenging ECV to measure in the UT/LS (Upper Troposphere/Lower Stratosphere) [14,15], which is a key zone of exchanges between the troposphere and the stratosphere [16] and should be monitored, especially within the tropics [17].

Water vapor measurement techniques are various—in situ or remote sensing, ground-based, airborne or space based [18]. Some ground-based Raman lidars are able to reach the UT/LS—the Purple Crow Lidar (Canada; [19]), the Table Mountain Facility Lidar (United-States; [20]), the Tor Vergata system (Italy; [21]) and two mobile lidars, ALVICE (Atmospheric Lidar for Validation, Interagency Collaboration and Education, in operation at the Howard University in Beltsville, United-States; [22,23]) and STROZ (STRatospheric OZone lidar; [24]) systems. The altitude of the tropopause is lower in mid-latitude areas than in the (sub)tropics. Raman lidars may be able to measure water vapor in the tropical UT/LS but such performances remain to be demonstrated and quantified precisely. The Mauna Loa Observatory Raman lidar can reach 15 km on a routine basis where the tropopause altitude is between 16 and 17.5 km asl depending on the season [25].

The calibration of Raman lidars can be calculated using two different methods. The first method consists of using a complementary water vapor measurement, typically radiosoundings [26–28], and the calibration therefore depends on the accuracy of another instrument that also presents its own limitations. Radiosondes are the most commonly used but other external sources can be used for calibrating the Raman lidars—GNSS (Global Navigation Satellite System) [29], microwave radiometer [30], sun photometer [31], independent lidar [32], kite-based humidity sensor [33], satellite and/or model data [34,35]. All fixed NDACC (Network for the Detection of Atmospheric Composition Change) Raman water vapor lidars use Vaisala sondes for the calibration method. The second approach consists of making an independent determination—different relevant parameters of the lidar system are measured or calculated experimentally [26], as for the ALVICE system, for example [36]. Regarding the dependent calibration, different ancillary measurements using the water vapor total column have been tested and compared to calibrate the data (GNSS, microwave radiometer and optimized matching methods when using radiosoundings) during different campaigns—DéMéVap (Développements Méthodologiques pour le sondage de la Vapeur d'eau dans l'atmosphère; [37]), MOHAVE (Measurements Of Humidity in the Atmosphere and Validation Experiments; [24]), and HOPE (HD(CP)$^2$ Observational Prototype Experiment; [30]). The uncertainty on the calibration factor calculated by the different instruments was between 5 and 10%. One conclusion of these studies is that the Raman lidar calibration can be achieved with the GNSS IWV. The first uncertainty estimates are around 7% during DéMéVap [37] and can be considered compatible with the scientific monitoring objective of the mid-upper troposphere.

One of the key restrictions when using the GNSS IWV for the calibration of Raman lidar water vapor measurements is the ability of the lidar to measure water vapor close enough to the ground. On Reunion Island, a new Raman lidar has been designed to simultaneously monitor the water vapor up to the lower stratosphere and the temperature from the stratosphere up to the thermosphere. The system is an updated version of the former Rayleigh-Mie-Raman system that was in operation at Saint-Denis (Reunion Island) between 2002 and 2010 [35], including a design allowing the retrieval of water vapor down to the ground to ensure a calibration based on GNSS IWV. In October 2012, the new lidar system was set up at a higher altitude at the Maïdo facility located at 2160 m asl [38]. At the same time, most of the remote sensing instruments of the OPAR (Observatoire de Physique de l'Atmosphère de la Réunion) were moved to the mountain observatory. In order to optimize the Raman lidar configuration and to evaluate a first set of data, the MALICCA-1 (MAïdo LIdar Calibration CAmpaign) campaign was organized in April 2013 [39]. The exploitation of the MALICCA-1 dataset [21] shows a relatively good agreement between the water vapor data measured by the lidar and those measured by the other instruments (Vaisala RS92, MLS; Microwave Limb Sounder). The relative difference

between 15 simultaneous measurements of the lidar and the RS92 sondes were lower than 10% for the lower and middle troposphere and between 10% and 20% for the upper troposphere. Two integration methods have been tested—240 min leading to an uncertainty of 2 ppmv between 17 and 20 km and a monthly integration (i.e., an integration of $\sim$ 1920 min) leading to an uncertainty of 1 ppmv at 20 km, which demonstrated the ability of the lidar to measure quantities of only a few ppmv in the UT/LS [40]. The monthly mean profile of water vapor based on MLS data agrees well with the mean lidar profile of MALICCA-1 in the lower stratosphere. The main conclusions of Dionisi et al. [40], based on two weeks of intensive measurements, needed to be reviewed using a longer period of routine measurements. In particular, a robust methodology for the long term calibration of the Maïdo Raman water vapor lidar (called hereafter "Lidar1200") using GNSS data had to be developed and validated.

The main objectives of the present study are to present this calibration method and to validate the first 2-year dataset of water vapor profiles. We will describe the Raman lidar and the database in Section 2. Ancillary measurements used for calibration and validation are introduced in Section 3. The data processing, including the calibration method, is detailed in Section 4. The performances of the instrument on a routine basis are evaluated in detail in Section 5. The validation of the profiles by comparing the lidar data with independent CFH radiosoundings is presented in Section 6.

## 2. Description of the System and the Database

### 2.1. The Raman System

Before being transferred to the Maïdo Observatory in 2012, the Raman water vapor lidar operated at sea level in the north of Reunion Island, at Saint-Denis [35]. Critical points have been handled (fluorescence, power and parallax effects) in order to optimize the new configuration of the system. The emitted wavelength was changed to 355 nm, which is more efficient than 532 nm [40]. Laser pulses are generated by two Quanta Ray Nd:Yag lasers with a repetition rate of 30 Hz, an energy of 375 mJ pulse$^{-1}$ and a duration of 9 ns. The two lasers are synchronized with a pulse generator with an uncertainty of less than 20 ns. The geometry for transmitter and receiver is coaxial for three reasons—(i) to avoid parallax effects; (ii) to extend the measurements down to a few meters from the ground and (iii) to facilitate the alignment. The overlap, which depends on the laser emission, the field of view (FOV) of the telescope and the optical unit is partial from the ground (i.e., 2.2 km asl) to 4 km asl. The backscattered signal is collected by a Newtonian telescope with a primary mirror of 1200 mm diameter.

It has been shown that the fluorescence in optical fibers can cause systematic biases [41]. Thus, no optical fiber is employed—a separation unit is used directly after the telescope to separate the Raman and Rayleigh signals. The FOV of the system is adjustable (from 3.0 to 0.5 mrad) thanks to a diaphragm field stop at the entrance of the separation unit. A 2 mm FOV (resulting in a 0.5 mrad FOV) allows the background light to be reduced and limits photon counting saturation from low altitude scattering, but results in a higher total overlap height. Details on the different tests run during MALICCA-1 that led to the final choice of the operational optical configuration are given in Dionisi et al. [40]. The separation unit is composed of dichroic beam splitters and interference filters, which separate the backscattered light. Hamamatsu miniature PMT (photomultipliers tubes) and Licel transient recorders are used for the photon detection and data acquisition (in photon counting only). The dead time value of the detection system given by the manufacturer is 3.7 ns. The raw data correspond to the integration of the signal over 1 min.

### 2.2. The Lidar Water Vapor Dataset

During a test phase (from October 2012 to October 2013), specific technical studies were conducted. Since November 2013, the instrument has been operated on a routine basis. The operation of the instrument became more stable after November 2013. The first two years of data (from November 2013 to October 2015) are used to set up a measurement routine, to develop a calibration methodology based on the use of GNSS data and to evaluate the instrument's performance based on instrumental

comparisons. The calibration method applies to these two years (with the method described in Section 4). During this period, the number of nights of measurement increased from 66 to 84 nights per year. The number of hours of operation increased by almost 60% year-to-year, from 185 h during November 2013–October 2014 to 293 h during November 2014–October 2015. This dataset includes the routine measurement periods but also intensive observation periods during campaigns. The time slot of routine operations is 19:00 to 01:00 +1 LT (i.e., 15:00 to 21:00 UTC). Efforts to increase the duration of the measurement sessions during campaigns ensured that 40% of the measurement sessions in the second year were longer than or equal to 240 min. The measurements are subject to instrument problems and depend on the availability of the technical staff and, in addition, on meteorological conditions (clear sky nights only). Summer (December-January-February, DJF) is the rainy season with the lowest number of measurements. The frequencies of measurements for the other three seasons are similar, although it should be noted that statistics for the March-April-May (MAM) and September-October-November (SON) periods were boosted by the campaigns in November 2013, June 2014, November 2014, and May and September 2015.

## 3. Ancillary Measurements

### 3.1. Lamp Measurement and Logbook for the Instrumental Stability Survey

At the beginning of each night of measurement, a systematic lamp measurement is made in order to detect potential sudden instrumental changes in the reception system. The lamp measurement consists of lighting up the telescope with white light, which illuminates the sensors, and recording the "white noise" then obtained. This noise is averaged over the whole altitude range and for all the channels. Finally, the lamp measurement value corresponds to the ratio of Raman $H_2O$ on $N_2$ channel signals. This value is independent of the altitude and of the laser power; the losses are only due to the reception optics and to the efficiency of the sensors. For our system, a variation of this value by a factor (of at least) 2 was considered as an indicator of an instrumental change. The lamp measurement is not used directly for estimating the calibration coefficient but is very useful to reveal any change occurring on the reception part of the system. In addition, Whiteman et al. [42] have shown that this technique needs to be considered carefully as the lamp measurement does not detect instrument modifications on the emission part of the system such as changes or realignments of the optical components and changes on the lasers. Therefore, in addition to the lamp measurements, the logbook can be checked to reduce non-automatic detections of such instrumental changes [40].

### 3.2. GNSS Data for Calibration

The GNSS TRIMBLE-NETR9 (MAIG, http://rgp.ign.fr/STATIONS/#MAIG) receiver is in operation at the Maïdo Observatory since 2013 in order to record vertical total columns of water vapor (IWV) strictly collocated with the lidar. To determine the IWV from GNSS data, the atmospheric zenith total delay (ZTD) is estimated using the data processing software Gipsy-Oasis II v6.4, in PPP (Precise Point Positioning) mode [43]. Using surface pressure information, the ZTD can be divided into a hydrostatic term, that is, the Zenith Hydrostatic Delay (ZHD or so-called dry delay) calculated through the Saastamoinen formula [44] and the Zenith Wet Delay (ZWD, so-called wet delay). The wet delay is the propagation delay experienced by GPS signals due mainly to water vapor abundance. ZWD is converted into IWV, using surface temperature and empirical formulas [45,46]. The typical cutoff elevation angle is fixed at 7° to ensure that the area sounded for water vapor is as local as possible. The surface pressure and temperature come from a meteorological station collocated with the lidar. The measurement of the meteorological station is acquired every 4 s, then we average it over 5 min in order to have the same temporal resolution as the GNSS data processing. There are some gaps in the GNSS IWV database over the two years of measurements, which represents about 23% of the nights of lidar measurement. This lack of data is partly due to GNSS and weather stations' availability, with operating rates of 93% and 77%, respectively. It was therefore necessary for us to set

up a calibration method that would allow us to calibrate all lidar data even in the absence of GNSS data. The GNSS IWV uncertainty results from the uncertainty on the ZTD, provided by Gipsy-Oasis, as well as from the uncertainty of the pressure and temperature sensors of the meteorological station (determined according to the sensor's datasheet) [47]. The total uncertainty ranges between 0.8 mm and 0.9 mm, depending on the GNSS IWV value. This value is consistent with that found in the literature—about 1 mm [37,48].

### 3.3. CFH Soundings for Validation

The frost point hygrometer sondes are widely recognized as being among the most accurate way to measure relative humidity in the UT/LS [49,50]. In 2010, after a comparison of different capacitive sensor radiosondes, the last WMO (World Meteorological Organization) report [51] concluded that only CFH sondes were able to measure the water vapor in the stratosphere. The CFH is a light balloon-borne hygrometer. The uncertainty of the CFH sonde is less than 4% in the lower tropical troposphere, less than 9% in the area of the tropopause and around 10% in the middle stratosphere around 28 km [49]. For the best measurements, the mixing ratio uncertainty can even reach 1% in the lower troposphere and 2% to 3% in the stratosphere [52].

Six CFH sondes were launched at the Maïdo Observatory during the MORGANE campaign. One of them was launched during the daytime and has not been included in the following comparisons. The main goals of MORGANE were—(i) blind comparisons between the temperature, water vapor and ozone lidars of the observatory and the NASA/GSFC (Goddard Space Flight Center) STROZ mobile lidar [53] brought to Reunion Island for the campaign (the results of these comparisons will be the subject of a dedicated paper); (ii) daytime and night-time meteorological radiosoundings to work on technical issues associated with GRUAN (Global Climate-Observing Systems GCOS Reference Upper Air Network) and (iii) the study of atmospheric processes associated with the composition of the UT/LS and with atmospheric dynamics. Almost 221 h of acquisition were collected by the lidars (ozone and water vapor) of the Maïdo Observatory between 11 and 22 May 2015 to evaluate the performances of the lidars. In the present study, the water vapor Raman lidar data were compared with the CFH dataset for validation.

## 4. Lidar1200 Water Vapor Profile Retrieval

### 4.1. Water Vapor Mixing Ratio Retrieval Equation, Sources of Uncertainties and Digital Filtering

As a reminder, with a 355-nm emitted wavelength, we use the 387 nm ($N_2$) and 407 nm ($H_2O$) Raman shifted wavelengths to retrieve the water vapor mixing ratio. The initial data processing algorithm [40] has been upgraded to include identification and quantification of the uncertainties and determination of the vertical resolution. The equation to retrieve the water vapor mixing ratio ($WVMR$), based on Whiteman et al. [54], can be rewritten as follows:

$$WVMR\,(z) = C\,O\,(z)\,F\,(z)\,\frac{P_{H_2O}(z) - b_{H_2O}(z)}{P_{N_2}(z) - b_{N_2}(z)}\Delta\tau, \tag{1}$$

where $C$ is the calibration coefficient, $O(z)$ is the overlap factor, $F(z)$ is the temperature dependence of the Raman cross-section, $P_i(z)$ is the number of photons received by the detector for $i$ (with $i = H_2O$ or $i = N_2$), $b_i(z)$ is the sky background (including the digital offset in the detector chain), and $\Delta\tau$ is the differential term due to the atmosphere.

The number of photons received by the detector might not correspond to the real number of photons, in particular when this number is high and the detector is saturated. For a non-paralyzable photon counting system, the signal can be desaturated according to the following equation, based on Müller [55]:

$$P_i = \frac{R_i}{1 - \tau_i \frac{c}{2\delta zL} R_i}, \tag{2}$$

where $R$ is the number of photons counted by the detector, $\tau_i$ the dead-time parameter of the detector, $c$ the speed of light, $\delta z$ the space resolution, and $L$ the number of shots of the laser.

The different sources of the statistical uncertainty are associated with the counting of the number of photons collected by the detector for the water vapor and nitrogen channels (depending on the altitude). The main sources of systematic uncertainties are the determination of the calibration coefficient, the temperature-dependence of Raman backscattering, overlap function ratios and the differential transmission in the atmosphere at the wavelengths of the water vapor and nitrogen Raman channels. Each of these uncertainties has been calculated or estimated (details are given in Appendix A). The uncertainties on saturation correction, fluorescence and differential transmission due to aerosols are neglected [40]. The uncertainty depends strongly on the integration time and the filtering of the signal. Thus, it is important to use a suitable digital filter regarding the vertical resolution and the order of magnitude of the total uncertainty.

The raw vertical resolution is 15 m. Data are smoothed with a low-pass filter using a Blackman window; thus, the final resolution is different from the initial resolution. The number of points of the digital filter increases with the altitude to compensate for the signal-to-noise ratio (SNR) decrease. The NDACC has formulated and adopted two standardized definitions for the calculation of the final vertical resolution for its lidars, based on (1) cut-off frequency of digital filters (used here) or (2) the full-width at half-maximum of a finite impulse response [56]. Based on the number of points used for the filter to vertically average the data, the vertical resolutions that can be derived from these definitions are 100–200 m in the lowest layers, 500 m in the mid-troposphere, 600 m in the upper troposphere and 700–750 m in the lower stratosphere, for a digital filter using the Blackman coefficients reaching 121 points at 20 km asl.

To summarize, the profiles are processed with a vertical resolution and an integration time, both depending on the water vapor variability at several levels. In order to convert the backscattered radiation profiles into water vapor mixing ratio profiles, it is necessary to calculate a calibration coefficient from water vapor column ancillary data afterward. The specific calibration method that has been developed for the Lidar1200 is described in the next subsection. The validation and evaluation of the performances of the lidar is detailed in Sections 5 and 6.

### 4.2. Calibration Method

#### 4.2.1. The GNSS Technique

Two ancillary measurements could be used to calibrate the Lidar1200: radiosondes and GNSS. The main concern for the calibration is to be able to have collocated, simultaneous measurements on a routine basis. The daily 12:00 UTC radiosounding (Météo-France) and the weekly 10:00 UTC ozone radiosounding (a NDACC, SHADOZ /Southern Hemisphere ADditional Ozonesondes and OPAR site) are performed at the airport, which is 20 km from the Maïdo Observatory. Then we consider that the space-time collocation criteria with the biweekly normal operation of the Raman lidar (15:00–21:00 UTC) at the Maïdo Observatory is not fulfilled. Launching two sondes per week from the Maïdo Observatory has a significant financial and logistical cost. On the other hand, GNSS measurements are abundant, collocated, independent and simultaneous with the lidar data. These are the main reasons for choosing the GNSS for the calibration. The use of GNSS measurements avoids the risk of changes being made to certain instruments by the manufacturer as has occurred with radiosondes in the past. Since 2013, the lidar profiles are calibrated with integrated water vapor columns obtained from GNSS measurements to ensure a better stability of the calibration in the long-term.

The main prerequisite to the use of GNSS IWV to calibrate Raman lidar water vapor measurements is the ability of the lidar to actually probe the IWV, including starting measurements at ground level. The emission and reception parts being coaxial, the useful signal starts right above the telescope. Figure 1 shows the dispersion of the first lidar measurement points (an integration of water vapor

from the telescope to 15 m above, over 5 min) with the collocated FTIR (Fourier-Transform InfraRed spectroscopy) meteorological station humidity measurements (at around 2 m above the telescope, 5-min average) between 2013 and 2015. The pressure and temperature of the latter are used to convert the water vapor data obtained by the lidar in g kg$^{-1}$ into relative humidity as a percentage. The linear regression of the data line is close to the 1:1 relationship. It should be noted that there is difference reaching 9% of humidity (when the humidity is lower than 30%) between the lidar and the FTIR station's measurement, and reaching 4% when it is higher than 80%. This could be explained by the difference in measurement techniques (a single point of measurement versus a vertical average). Figure 1 shows that there is a good overall correlation between measurement of the lidar and the FTIR station.

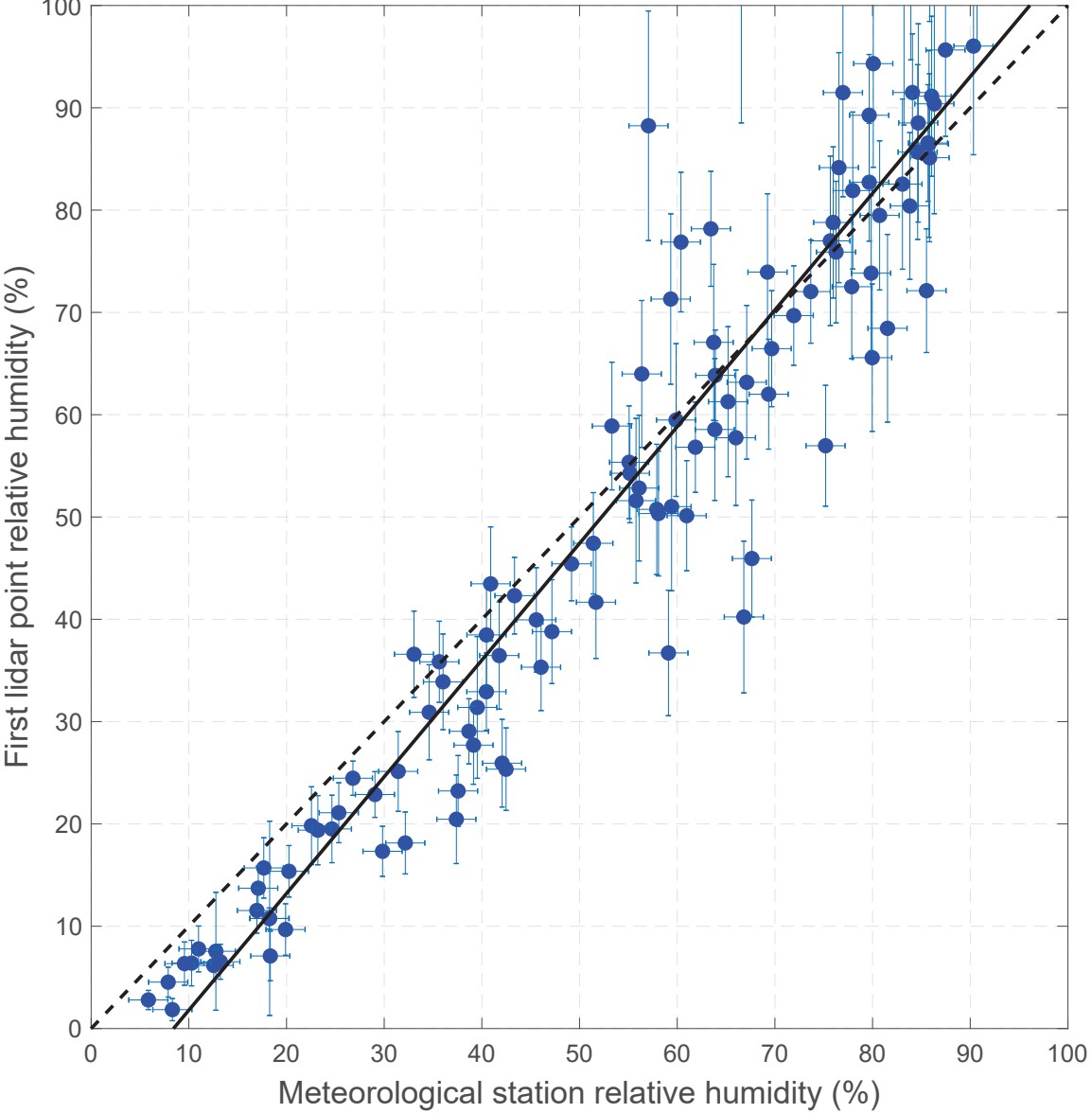

**Figure 1.** Dispersion of the relative humidity (%) between the first vertical point of the Lidar1200 and the Fourier transform infrared (FTIR) meteorological station (Maïdo Observatory) between November 2013 and October 2015. The black dotted line shows the 1:1 relationship and the black dashed line shows the best linear fit to the data. The uncertainty of each instrument is indicated with a blue line, vertical for the total uncertainty of the lidar and horizontal for the 2% uncertainty of the meteorological station.

Another concern regarding the use of total column is that, sometimes, the lidar profile could not reach the lower stratosphere and thus concerned only a partial column. A calculation performed on the vertically averaged water vapor data of the CFH sondes launched during the MORGANE campaign (Reunion Island, May 2015), shows that, up to 5 km, the cumulated water vapor represents 90% of the total column; above 10 km, 99% of the whole column is contained. Thus, it appears sufficient to use IWV to calibrate the Lidar1200 water vapor profiles when the range of the profile is higher than 10 km, which is almost always the case for a 5-min integration.

### 4.2.2. Description of the Method

Figure 2 describes the calibration method developed for the Lidar1200. Usually a calibration coefficient is the ratio between a reference instrument, the GNSS IWV here and the uncalibrated lidar IWV data. To convert the lidar profile (in g kg$^{-1}$) into IWV (in mm), the lidar data are vertically integrated in function of the pressure as explained in Bock et al. [57]. The pressure profile that we use is extracted from an AERIS (https://www.aeris-data.fr/) product called Arletty, which is computed on demand by spatial linear interpolation in the grid of the ECMWF (European Centre for Medium-Range Wheater Forecasts) analysis at the specific location of the Maïdo Observatory. In this section, the 5-min calibration coefficient—named "5-min coefficient"—represents the ratio between the GNSS IWV calculated every 5 min and the lidar IWV integrated over the same 5-min window. The 5-min coefficient varies in time, probably because the integration methods are different. Even if the GNSS and the lidar are collocated, they do not measure exactly the same volume of water vapor (in space). With hindsight on the dataset, a manual identification of the periods' change of calibration coefficient can be made by checking the results of the lamp measurements and the logbook overview. These 5-min coefficients are averaged each night of measurement and are named "nightly coefficient". The average nightly coefficient between two instrumental changes detected as described above is considered as the "calibration coefficient" of each measurement performed during this period. The acquisition time of the lidar changes from one night to another, that is the reason why we chose to take the average of the nightly coefficients rather than the average of the 5-min coefficients of each period. If we took the 5-min coefficient average, the longest session of measurements would tilt the average and thus the final calibration coefficients. To conclude, at the end of the process, there is only one calibration coefficient for all the data belonging to a same period, regardless of the integration time of the lidar measurement. This means that the measurement is re-calibrated only after an instrumental change. For some data, the GNSS IWV cannot be calculated because of missing pressure and temperature data. The associated lidar measurements can still be calibrated by using the predetermined calibration coefficient of the period delimited by two instrumental changes framing the lidar measurement.

The uncertainty of the calibration coefficient of each period can be estimated by the standard deviation of the nightly coefficients of the associated period. It should be noted that, given that the variability of the calibration transfer coefficient is influenced by atmospheric variability, the real uncertainty is probably somewhere between the standard deviation of the mean and the standard deviation. We assume that during a given period, the calibration coefficient remains stable and that the factors explaining fluctuations of the calibration coefficient are due to measurement uncertainties on both lidar and GNSS and also atmospheric variability through random atmospheric conditions and random line of path observations. Two uncertainties have to be taken into account regarding the calibration—the uncertainty in the transfer of the calibration from the GNSS to the lidar and the uncertainty of the external source of calibration, that is, the GNSS IWV described in Section 3.2. The total uncertainty budget for the lidar data can be found in Appendix A.

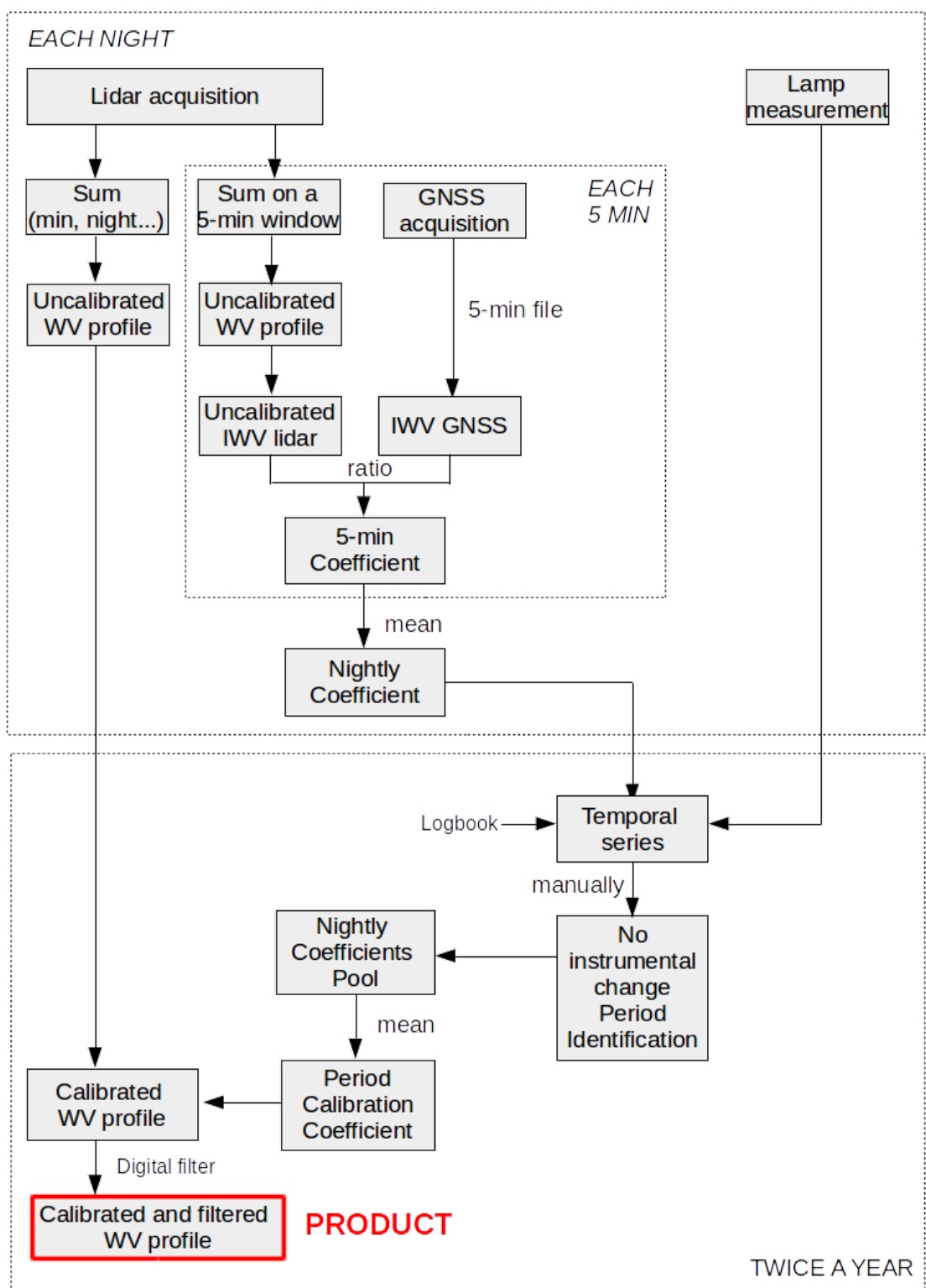

**Figure 2.** Calibration procedure of Lidar1200's water vapor profiles. WV stands for Water Vapor.

### 4.2.3. Application over 2 Years of Data

Figure 3 shows the time series of the nightly coefficients (blue dots) of the Lidar1200 water vapor measurements between November 2013 and October 2015 and the changes (indicated by the vertical dotted lines). The calibration coefficients of the 9 periods are listed in Table 1 (corresponding to the horizontal solid red lines in Figure 3). The raw water vapor profiles of one period are calibrated with

the average nightly coefficient of the period. That means, for example, that the water vapor profiles of 20 April 2015 are calibrated with a ratio of 203, as well as all the lidar profiles retrieved between 20 April and 11 May 2015. The measurements are re-calibrated for the change of period (from P07 to P08) on 12 May 2015, following a realignment of the optics of the lidar. The standard deviation of the nightly coefficient for each period varied between 6% and 10% between 2013 and 2015. The mean standard deviation of the dataset is 9%.

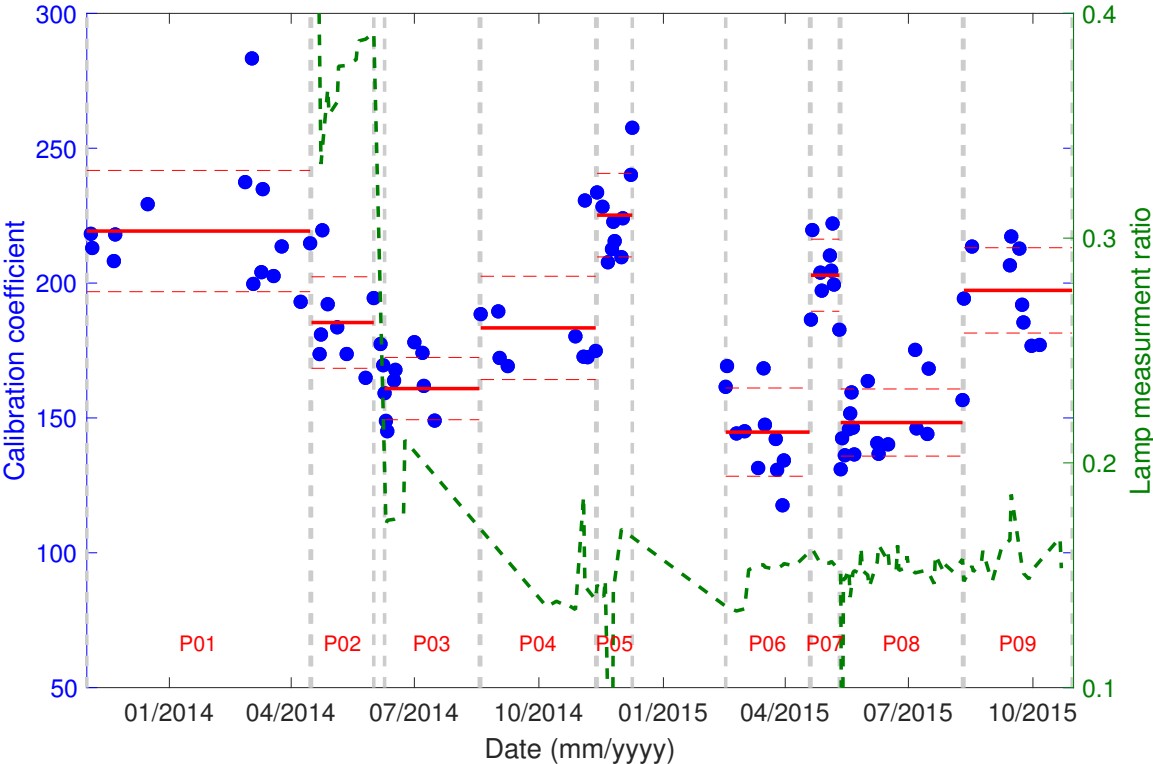

**Figure 3.** Time series of the nightly coefficients (blue dots) for the Lidar1200 2-year dataset. The vertical dotted lines separate the nine quasi-stationary periods (P01 to P09) of the calibration coefficient. Horizontal solid red segments display values of the average nightly coefficients over the corresponding periods, i.e., the calibration coefficients. The horizontal dotted red segments correspond to values of the mean nightly coefficients plus or minus the standard deviation. The green dotted line corresponds to the lamp measurements made at the beginning of each measurement night from April 2014. The mm/yyyy dates correspond to the first day of the associated month.

**Table 1.** Chart recapitulating the different calibration coefficients of the water vapor data of the Lidar1200 between November 2013 and October 2015.

| Period | Dates (Day/Month/Year) | Calibration Coefficient | Standard Deviation |
|--------|------------------------|-------------------------|--------------------|
| P01 | 01/11/2013–15/04/2014 | 219 | 22 |
| P02 | 16/04/2014–01/06/2014 | 185 | 17 |
| P03 | 09/06/2014–18/08/2014 | 161 | 12 |
| P04 | 19/08/2014–12/11/2014 | 183 | 19 |
| P05 | 13/11/2014–09/12/2014 | 225 | 15 |
| P06 | 16/02/2015–19/04/2015 | 145 | 16 |
| P07 | 20/04/2015–11/05/2015 | 203 | 13 |
| P08 | 12/05/2015–10/08/2015 | 148 | 12 |
| P09 | 11/08/2015–30/10/2015 | 197 | 16 |

### 4.2.4. Comparison with Calibration by Radiosounding

The calibration method is now compared with other widely used calibration techniques, in particular the use of radiosoundings. Dionisi et al. [40] showed encouraging results for the use of GNSS IWV based on a first comparison with the use of Vaisala RS92 to calibrate profiles during the MALICCA-1 campaign. The MORGANE campaign involving CFH sondes launched from the Maïdo Observatory concomitantly with GNSS and lidar measurements provided an interesting opportunity to evaluate the robustness of the GNSS calibration method. The radiosonde-based calibration method was adapted from Whiteman et al. [23], which has been shown to reduce the influence of atmospheric variability on the calculation of the calibration coefficient. Within the height range of 3 to 13 km asl (for CFH sondes), the algorithm performs a linear regression between the lidar and the radiosonde profiles by segments of 0.6 km (30 points). Then, to improve this regression, the radiosonde profile is smoothed with the same filter than the lidar profile. Ordered pairs of data with a correlation value $R^2$ of more than 0.95 and with an accuracy (calculated with method of least mean squares) lower than 25% are selected to ensure that the lidar and the radiosonde are sampling the same atmospheric layer. The final dataset, composed of more than 60 points, was used to calculate the calibration coefficient as the mean of the ratio of the ordered pairs. During the MORGANE campaign, the $R^2$ and the accuracy ranged from 0.76 to 0.92 and from 8% to 24%, respectively. The final calibration value is the mean ratio between the data of each ordered pair. The time series of daily calibration coefficients between 15 and 22 May for the CFH and for the GNSS when the Lidar1200 was operating and when a radiosounding was performed simultaneously are shown in Table 2. The mean GNSS calibration coefficient for the selected dates is 146 ($\pm$10). The calibration coefficient based on the CFH sondes is 154 ($\pm$9) (Table 2). The calibration coefficient of the Lidar1200 shows a mean bias of around 5% compared to the calibration coefficients derived from the CFH technique (Table 2), this bias is clearly negative. The MORGANE campaign is included in the period n°8, for which the calibration coefficient is 148 ($\pm$12) (Table 1). The mean coefficient derived from the CFH is included in the standard deviation interval of the routine GNSS based calculation. The calibration coefficient of period n°8 shows a mean bias of less than 4% with the calibration coefficients derived from the CFH technique (Table 2). Taking the uncertainties of the different instruments into account, the two techniques are in an acceptable agreement, showing that the GNSS technique is as suitable as radiosoundings for the calibration of the water vapor profiles of the Lidar1200. Both techniques show a day-to-day variability (around 6% for the sondes and around 7% for the GNSS).

**Table 2.** Lidar1200's nightly (and mean) coefficients calculated with CFH and GNSS IWV between 15 and 22 May 2015.

| Date (Day/Month/Year) | Calibration Coefficient with CFH | Calibration Coefficient with GNSS |
|:---:|:---:|:---:|
| 15/05/2015 | 139 | 136 |
| 18/05/2015 | 154 | 146 |
| 19/05/2015 | 163 | 152 |
| 20/05/2015 | - | 160 |
| 21/05/2015 | 161 | 146 |
| 22/05/2015 | 154 | 137 |
| Mean | 154 | 146 |

## 5. Performances of the Lidar1200 in Monitoring Water Vapor on Routine Basis

The previous section detailed the data processing of the Lidar1200 profiles, with a focus on the calibration method. The following section aims to provide an overview of the mean performances on a routine basis.

*5.1. Mean Performances*

Three classes of mean profiles were calculated to evaluate a mean uncertainty that might be associated with the Lidar1200 water vapor data regarding the altitude and to help define an empirical user's guide for the data integration time leading to a compromise between scientific purpose, uncertainty and vertical resolution (Appendix B). All the 10, 40 and 240 min unfiltered and uncalibrated measurements were averaged. Those classes of profiles were processed with the operating algorithm, associated uncertainties were calculated and a suitable digital filter was applied (21, 61 and 121 points, respectively). Figure 4 represents the mean profile for each class, their total uncertainty and the vertical resolution associated. The average profile of 240 min is drier between 6 and 8 km asl than those of 10 and 40 min. This difference is explained by the fact that the sample size of the 240-min measurements is smaller than that of the 40 and 10-min measurements, as some nights of measurement did not reach 240 min between 2013 and 2015. The mean total uncertainty for the average profiles of 10, 40 and 240 min reaches a threshold of 20% at around 9.8, 12 and 14.2 km asl, respectively (Figure 4). These three classes of profiles provide information on the water vapor mixing ratio in the large part of the troposphere and offer various time scales for studying atmospheric processes. At 12 km altitude, they present a total uncertainty of 79%, 20% and 11%, respectively (Figure 4). The 240-min class reaches the tropical tropopause with an uncertainty of more than 80%. It appears to be important to reach the tropopause with a smaller uncertainty, and thus to integrate the measurements for longer periods.

Hoareau et al. [35] estimated the performances of the Lidar1200 for water vapor measurements based on numerical simulations. They showed that, with 30 min of integration, the statistical uncertainty would be 15% at 14.6 km and 30% at 16.3 km. Our study of the 40-min average profile for November 2013–October 2015 shows that the statistical (total) uncertainty reaches in fact 15% at 11.9 km (11.2 km) and 30% at 12.9 km (12.7 km). The simulation of performances by Hoareau et al. [35] was run for a clear sky, with no moon and using two wavelengths (355 and 532 nm). In practice, the Lidar1200 operates at only one wavelength and for different phases of the moon. The measurement of water vapor during 9 h in September 2015 showed that the moon phase had a non-negligible influence on the quality of the data (see Section 5.2). Using all routine measurement nights (potentially influenced by the moon and variable meteorological conditions) for a 240-min integration time, the Lidar1200 shows a total uncertainty of 15% at 13.5 km and 30% at 15.2 km (Figure 4).

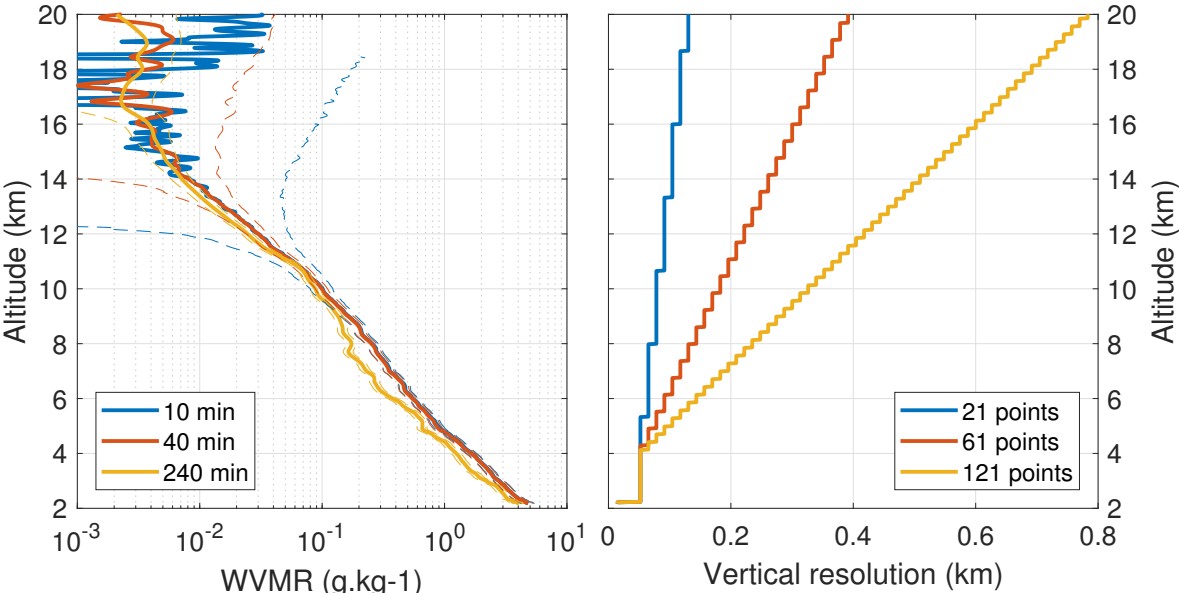

**Figure 4.** Mean profiles integrated over 10 min (solid blue line), 40 min (solid orange line) and 240 min (solid yellow line) with their total uncertainty (dotted lines) (**left**), and the associated vertical resolution (**right**) according to the altitude above sea level.

*5.2. Maximum Altitude Range*

One of the main objectives during the field campaigns was to determine the maximum altitude range that could be reached with a reasonable uncertainty (a maximum of 20%–30% depending on the scientific purposes). Excluding logistical issues associated with a manually operated lidar and assuming a night with clear sky, the measurement session would last ten hours at most. A test was performed during the LIDEOLE-2 campaign (September 2015, Maïdo Observatory) with a night session of about 9 h. The profile of 24 September 2015, integrated over the whole night, reaches approximately 15.4 km with an uncertainty of ∼30% (∼2.7 ppmv) (not shown). Although the sky was clear, the lidar performance was hampered by the presence of the full moon in the system field of view between 15:00 and 17:00 UTC—between the beginning and the end of the measurement session, the sky background on the 387 nm (407 nm) channel increased by a factor of 100 (10). Because of these limiting factors, we realize that this 9-h profile does not reach an altitude much higher than the average profile of 240 min (Figure 4) with a total uncertainty of 30%. The increase in background affected the first two hours of measurements, limiting the all-night profile to a maximum altitude of 16 km with a total uncertainty of 40%. With a lower background noise, this 9-h profile would probably have reached the lower stratosphere. We realize the difficulty of obtaining night-time measurements to reach the tropopause with a total uncertainty of 40%. This shows the need for the automation of the Lidar1200 system, which requires the intervention of operators. This would make it possible to extend the measuring ranges as much as possible and multiply the number of nights of measurement with favourable conditions (clear nights, new moon, etc.). The automation of lidar systems is a work in progress. Another way to obtain profiles reaching tropopause with a reasonable total uncertainty would be to integrate over several nights of measurement. Several studies used monthly integration to optimize the signal and produce water vapor profiles up to the UT/LS [21,40].

## 6. Validation of the Lidar1200 Dataset: Comparison with CFH Sonde Data during the MORGANE Campaign

In the following section, the water vapor measurements of the Lidar1200 are compared in the troposphere and stratosphere with those of the CFH sondes, a reference instrument.

*6.1. Methodology of Comparison between CFH and Lidar1200 Water Vapor Profiles*

A methodology of comparison of the data has been developed. For these comparisons, the lidar is calibrated with the GNSS IWV-based methodology described in Section 4.2.2. The CFH profiles are averaged each 15 meters (rolling average). The CFH uncertainty of each daily profile is based on the literature; 4% in the troposphere (applied to the 2.2–14 km range) and 9% around the tropopause (applied above 14 km). Then, the digital filter described in Section 4.1. is applied to the CFH profiles and their uncertainty. We chose to reconcile the resolution of the CFH with that of the lidar. For the comparisons in the troposphere, water vapor profiles retrieved by the radiosondes are compared to the 40-min sets of lidar data starting at the launch time of the balloon, except for the 15 May lidar data, for which there was a shift of 11 min between the lidar session and the balloon launch due to a delay in starting lidar operations. To compare the profiles in the UT/LS, the time integration for the lidar sessions needs to be expanded. Between 14 and 17 km, the relative difference is minimized if the whole night measurement session of the lidar is used. Above 17 km, the relative difference is minimum if the whole campaign measurements are compared with the mean CFH profile of the campaign. The variability over the 15 days of the campaign seen by the CFH is less than 0.5 ppmv between 17 and 22 km asl. Thus, the temporal variability of the water vapor at these altitudes allows an individual profile to be compared with the lidar profile at the scale of the whole campaign. To sum up, in the following, the Lidar1200 integration times used are adjusted with respect to the tropospheric columns analyzed: 40 min, night, and the whole campaign (∼48 h) for the 2.2–14 km, 14–17 km and 17–22 km partial columns, respectively. These integration times/probed partial column pairs were

established to maximize the lidar performances (vertical resolution, uncertainties) in these atmospheric layers following the considerations stated in Section 4.

*6.2. Results*

Figure 5 compares the Lidar1200 and CFH sonde mean water vapor mixing ratio profiles for the MORGANE campaign in the troposphere. The lidar data used for this comparison represents the average of the five 40-min coincident lidar sessions, while the CFH data is the average of the 5 CFH radiosoundings of the campaign. On the left, the mean CFH profile is superimposed on the mean lidar profile of the 40-min sessions in the lower and middle troposphere (Figure 5a). The relative difference between the lidar and CFH profiles (both smoothed with a digital filter using the Blackman coefficients reaching 61 points) is represented in Figure 5b. The CFH uncertainty in Figure 5 corresponds to the average of the uncertainties of the daily CFH profiles. The lidar uncertainty for Figure 5 corresponds to the average of the uncertainties of the 40-min lidar profiles. The mean relative difference (i.e., the vertical averaging of the absolute values of each altitude) for the 2.2–14 km partial column is 8.8%. The relative difference between the CFH and the lidar is characterized by dominantly being within the instrument uncertainties except for some peaks, at 3, 5.2, 8.5, 10.8 and 12.2 km asl (Figure 5b). All individual comparisons (not shown) agreed similarly for the peak at 3 km. The peak at 5 km asl is explained by a persistent sharp water vapor gradient, the altitude of which changes slightly from one day to another and that is seen differently in altitude because of the specific vertical resolution and digital filtering of the data of each instrument. During the night of 19 May, thin water vapor layers observed by the CFH sonde were different (in altitude or in amplitude) from those measured by the Lidar1200 which influence the relative difference shown on Figure 5b. The large spatiotemporal variability of the water vapor is our best explanation for the large differences observed on this particular night, and that does not call into question either the quality of the measurement or the methodology of comparison.

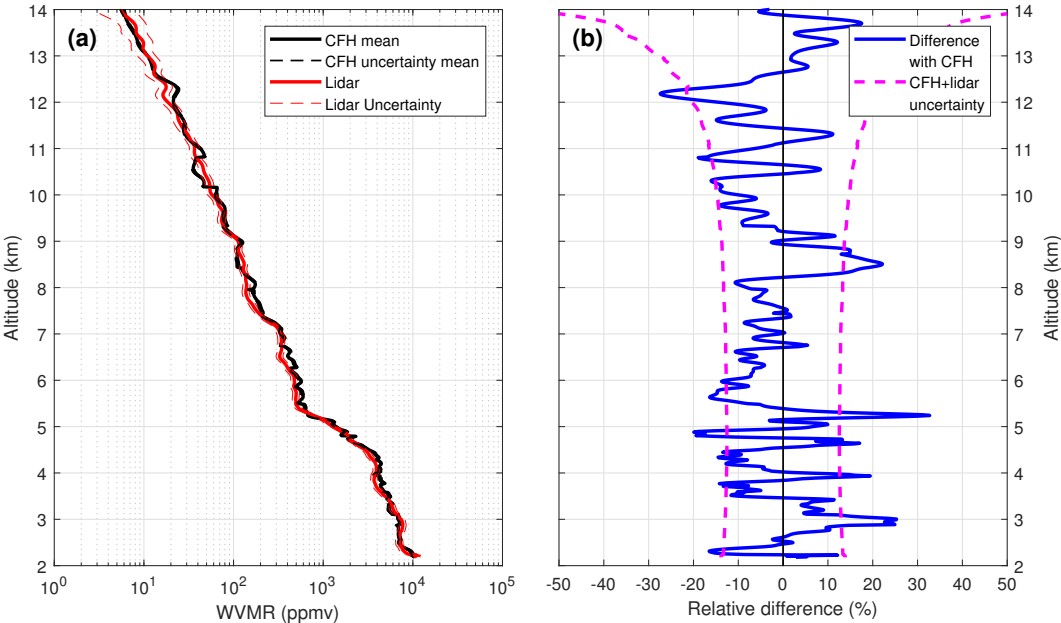

**Figure 5.** Comparison of the lidar and CFH water vapor mixing ratio during MORGANE in the troposphere between 2.2 km and 14 km. On the left (**a**): superimposition of the mean CFH water vapor profile (thick solid black line) with the lidar profile (solid red line), i.e., the mean profile of the 40-min sessions. The red dotted line represents the $\pm 1\sigma$ lidar total uncertainty and the black dotted line represents the CFH uncertainty. On the right (**b**): mean relative difference (in %) between the lidar and CFH profiles. CFH data are taken as reference. The sum of CFH and lidar uncertainties is indicated with the magenta dotted lines.

Figure 6 compares the Lidar1200 and CFH sondes mean water vapor mixing ratio profiles for the MORGANE campaign in the UT/LS. In the lower stratosphere (Figure 6a,b), the lidar data used for the comparison represent the integration of the whole 48-h dataset of the campaign. In the upper troposphere (Figure 6c,d), the lidar data used for the comparison represent the average of the 5 night sessions coincident with the CFH launches during the campaign. On the left, the mean CFH profile is superimposed on the lidar profile integrated over the whole campaign data in the lower stratosphere (Figure 6a) and on the mean lidar profile of the whole night sessions in the upper troposphere (Figure 6c). The relative difference between the lidar and CFH profiles (both smoothed with a digital filter using the Blackman coefficients) is represented on Figure 6b (with the filter reaching 121 points) and Figure 6d (with the filter reaching 201 points). The CFH uncertainty in Figure 6 corresponds to the average of the uncertainties of the daily CFH profiles. The lidar uncertainty for Figure 6c,d corresponds to the average of the uncertainties of the nightly profiles. There is good overall agreement between CFH and Lidar1200 between 14 and 22 km considering the uncertainties of both instruments (Figure 6). The mean relative difference (i.e., the vertical averaging of the absolute values of each altitude) between the mean CFH and Lidar1200 profiles is 8.6% in the 14–17 km layer (Figure 6d) and 5.5% in the 17–22 km layer (Figure 6b), which corresponds to less than 1 ppmv. The comparison shows a peak of around 18.5% at 15.5 km asl. The durations of the different nightly sessions were not equal, and this fact might have induced significant differences. Despite this peak of the difference, the mean CFH profile is included within the lidar error bar (Figure 6d).

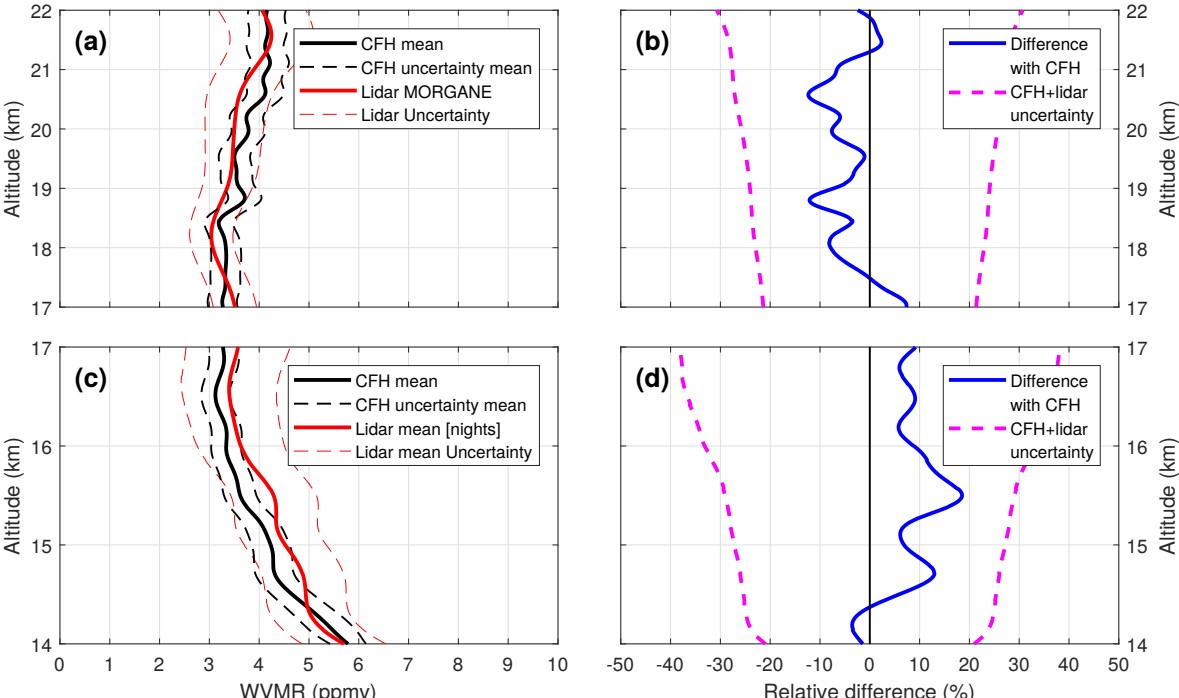

**Figure 6.** Comparison of the lidar and CFH water vapor mixing ratio during MORGANE in the UT/LS, between 14 km and 22 km. On the left (**a–c**): superimposition of the mean CFH water vapor profile (thick solid black line) with the lidar profile (solid red line). At the bottom left (**c**): mean lidar profile of the night sessions between 14 and 17 km asl and, at the top left (**a**): the lidar profile integrated over the whole campaign data between 17 and 22 km asl. The red dotted line represents the $\pm 1\sigma$ lidar total uncertainty and the black dotted line represents the CFH uncertainty. On the right (**b–d**): relative difference (in %) between the lidar and the mean CFH profile, between 14 and 17 km (**d**) and between 17 and 22 km (**b**). CFH data are taken as reference. The sum of CFH and lidar uncertainties is indicated with the magenta dotted lines.

To conclude, the Lidar1200 and the CFH profiles of the MORGANE campaign are in agreement from the ground to 22 km asl, except for some specific altitudes which could be explained by the spatio-temporal variability of the water vapor for the location of both instruments/for the air masses sampling. The average difference between the CFH and Lidar1200 of less than 9% suggests that the uncertainty that we have calculated (Figure 4) on the basis of our uncertainty budget in Appendix A most probably overestimates the true error. The good agreement between the average CFH profile of the MORGANE campaign and the integration of 48 h of lidar measurements in May 2015 shows the relevance of the monthly integration of Lidar1200 data. This approach will be the subject of a future study.

## 7. Conclusions

Since 2013, a Raman lidar has been operated at the Maïdo Observatory (Reunion Island). The latitude of Reunion Island (21°S) makes it possible to study tropical and subtropical mechanisms. Because of the large migration in latitude of the Inter-Tropical Convergence Zone that affects the atmosphere above the South-West Indian Ocean, the water vapor content might show both large and weak total columns, depending on the season. The calibration process needs a special attention when significantly dry upper tropospheric conditions are encountered, not only during austral winter. To face this challenge, a calibration methodology based on GNSS IWV for long-term stable data processing has been set up and a strategy of data sampling has been deployed to assess the performances of the Lidar1200.

At the Maïdo Observatory, the GNSS technique is used on a routine basis. A methodology based on the identification of instrumental changes to recalculate the calibration coefficient has been applied and tested on the 2013–2015 dataset. A comparative methodology has been performed using radiosondes launched simultaneously to lidar measurements. The calibration technique based on GNSS is within a 5% agreement with the mean calibration coefficient calculated by CFH sondes. The comparative results illustrate the relevance in using a GNSS-based calibration. The Lidar1200 mean profiles of 10, 40 and 240 min illustrate the total uncertainties in the troposphere. Layers with thicknesses of several hundred meters can be detected with a temporal resolution of 10 min up to around 10 km asl with less than 20% uncertainty. The 240-min average profile reaches 14 km asl with a total uncertainty of 20%, that is, a night of measurement allows information to be retrieved up to the upper troposphere (subject to good meteorological conditions such as a clear sky). This evaluation work has led to the creation of a user's guide—the definition of criteria for integration times and filtering adjusted to the process studied in terms of altitude, thickness and temporal scale. During the MORGANE experiment, the comparison between the results given by CFH sondes launched and the lidar profiles shows that both datasets are in agreement in the troposphere considering the uncertainties of both instruments. In the UT and in the LS, the absolute difference is lower than 1 ppmv. With respect to the CFH and lidar uncertainties, the instruments are in agreement up to the UT/LS and the CFH profiles validate the lidar data up to 22 km asl. These results show the possibility of integrating lidar measurements over several nights to obtain water vapor measurements in the UT/LS on a monthly scale for example. This would be further investigated in a next study. Thus the Lidar1200 provides water vapor profiles with high vertical and temporal resolution in the troposphere on a routine basis and is able to detect water vapor from the troposphere up to the UT/LS. A regular effort is made to perform instrumental comparisons through the recurrent CFH soundings performed at the Maïdo Observatory.

Further optimizations of the system are planned to improve the performance of the Lidar1200, especially by upgrading some optical components and cooling the PMT. The technology improvements in the detector might lead to a higher efficiency. The stability of the calibration method should be enhanced by using a larger dataset of GNSS IWV. Finally, adaptive or Optimal Estimation Method (OEM) algorithms could be used in an operational way in order to reveal fine-scale structures ([23,58] respectively) and refine the calculation of the uncertainties and of the calibration coefficient. The use

of such a method is part of the future development of the Lidar1200 water vapor data processing intended to bring it closer to the GRUAN (Global Climate Observing System-GCOS Reference Upper Air Network) requirements [59].

The abilities of the Lidar1200 has already been successfully used to study atmospheric processes such as stratosphere-troposphere exchanges [39,60]. The data of the Lidar1200, used alone and with other measurements will bring new results on the characterization of water vapor in the troposphere, the monitoring of the water vapor in the UT/LS, the study of the exchange processes between the stratosphere and the troposphere and the validation of satellite measurements.

**Author Contributions:** Conceptualization, P.K., V.D. and H.V.; methodology, P.K., G.P., H.V., V.D. and J.-L.B.; software, G.P.; validation, H.V. and G.P.; formal analysis, G.P. and H.V.; investigation, H.V., G.P., P.K., V.D. and J.-L.B.; resources, P.K., V.D., J.-P.C.; data curation, G.P., S.E., P.B., F.P. and S.K.; writing–original draft preparation, H.V.; writing–review and editing, G.P., H.V., V.D., J.-L.B., P.K., J.-P.C. and P.B.; visualization, G.P. and H.V.; supervision, P.K. and V.D.; project administration, J.-P.C.; funding acquisition, J.-P.C., P.K. and V.D.

**Funding:** We acknowledge the ACTRIS project and the support of the European Community (Research Infrastructure Action under the FP7 "Capacities" specific program for Integrating Activities, ACTRIS Grant Agreement no. 262254). The European Commission (FEDER program), Région Réunion and CNRS are acknowledged for their strong support in the building of the Maïdo facility. We are also grateful to Université de La Réunion and CNRS for their strong support of the OPAR station (Observatoire de Physique de l'Atmosphère de La Réunion, UMR8105, UMR8190 and UMS3365) and the OSU-R activities. The research leading to these results was supported by the French LEFE INSU-CNRS Program under the projects VAPEURDO and VEGA. We thank the CNRS (Centre National de Recherche Scientifique/France) and the University of Reunion Island for funding the Ph.D. of H. Vérèmes and the MALICCA campaigns.

**Acknowledgments:** The Lidar1200 data used in this publication were obtained as part of the NDACC and a level 2 product of daily vertical water vapor profiles will be publicly available through the NDACC portal (http://www.ndacc.org) and the French atmospheric data portal (https://www.aeris-data.fr/). The raw data and other products of the Lidar1200 water vapor data are available at the OPAR web portal (https://opar.univ-reunion.fr/), please contact us or osureunion-informatique@univ-reunion.fr to obtain data download login/password. The ZTD data used to compute the GNSS IWV are archived by the IGN on the ICARE data center (http://www.icare.univ-lille1.fr/). The authors gratefully acknowledge Eric Golubic, Patrick Hernandez and Louis Mottet who are deeply involved in the routine lidar observations at OPAR. Thanks are also due to Yann Courcoux, Jacques Porteneuve, Alain Hauchecorne, Franck Gabarrot, Jean-Marc Metzger, Nicolas Marquestaut, Jimmy Leclair de Bellevue, Aline Peltier and Patrice Boissier. We are also grateful to Davide Dionisi for his work on the water vapor data processing. We thank Holger Vömel and Ruud Dirksen for the processing of the CFH data. Christian Hermans and Martine De Mazière are acknowledged for the provision of the Reunion Island ICOS (Integrated Carbon Observation System) meteorological station data.

**Conflicts of Interest:** The authors declare no conflict of interest. The funders had no role in the design of the study; in the collection, analyses, or interpretation of data; in the writing of the manuscript, or in the decision to publish the results.

## Appendix A. Uncertainty Budget

The total uncertainty budget of the Lidar1200 water vapor profiles includes the following uncertainties:

- Uncertainty due to the detectors (PMT), which is a statistical uncertainty and follows a Poisson distribution. It is calculated by the square root of the signal. The ratio between this uncertainty and the signal increases with the altitude and depends on the digital filter used.
- Uncertainty of the background noise, which is calculated using the least squares method. It corresponds to the standard deviation of the background noise divided by the square root of the number of points of the signal used to calculate the background noise. In our case, the background noise is calculated as the mean of the signal between two altitudes.
- Uncertainty on the differential absorption, which is driven by the uncertainty on the extinction of the molecules. The atmospheric density profile is calculated with a model of atmospheric density having an arbitrary uncertainty fixed at 15% on this profile. After propagation of the uncertainty, it represents a negligible value of only 0.05% on the data at 20 km asl.
- Uncertainty due to the temperature-dependence of the Raman cross-sections, which is estimated to be maximum at the tropopause, at 6.7% [61].

- Uncertainty due to the overlap factor is estimated to be 4% at the ground and to decrease up to the maximum recovery altitude of the signal.
- Uncertainty on the calibration, which is a combination of the uncertainties on the external source of measurement and the transfer of the calibration source to the lidar profile. We estimate the uncertainty on the calibration coefficient to be the standard deviation of the nightly coefficients of a period and is of 9% in average.

## Appendix B. User's Guide

With the benefit of hindsight on the 2-year dataset, it is possible to define an empirical user's guide for the data integration time that leads to a compromise between uncertainty and vertical resolution (Table A1). Depending on the scientific investigations, specific filter points and integration times can be chosen. The vertical resolution is below 100 m for profiles of a 1 to 10 min (with the number of points of the filter reaching 21). Fine-scale structures (vertical thickness of several hundred meters) can be detected with 10-min integration profiles in the lower and middle troposphere. To perform an instrumental comparison in the troposphere with, for example, a radiosonde, an integration time of 40 min is necessary. To reach the upper troposphere (above 14 km asl), the integration time should cover at least 4 h. The total uncertainty proposed in Table A1 is based on all the sources of uncertainty described in Appendix A. The results of the instrumental comparisons (see Section 6) suggest that our calculation overestimates the uncertainty.

**Table A1.** User's guide for the Lidar1200 water vapor data treatment regarding the altitude range and current associated uncertainties. It corresponds to total uncertainties on the mean signals for 10-, 40- and 240-min integration (Figure 4).

| Altitude Range (km asl) | Temporal Resolution (min) | Vertical Resolution (m) | Total Uncertainty (%) |
|---|---|---|---|
| 2.2–10 | 10 | 65–90 | <20 |
| 2.2–12 | 40 | 100–300 | <20 |
| 2.2–15 | 240 | 100–650 | <30 |

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
