# Peer review of "Validation of the Water Vapor Profiles of the Raman Lidar at the Maïdo Observatory (Reunion Island) Calibrated with Global Navigation Satellite System Integrated Water Vapor"

_atmosphere, doi:10.3390/atmos10110713_

Round 1

Reviewer 1 Report

In this study, the authors are presenting their effort towards the calibration of the water vapor Raman lidar operating at the high-altitude observatory located in Reunion island. For this purpose, they used integrated water vapor data set obtained from the Global Navigation Satellite System, as well as range resolved meteorological profiles obtained by radiosondes. The manuscript is well written and holds scientific merit. Therefore I would kindly suggest the manuscript to be considered for publication in the Journal of Atmosphere of MDPI. In order to improve the quality of the manuscript I would kindly suggest to the authors to take into consideration the following minor comments: 

Line 12: A general comment for the manuscript is related to the water vapor accuracy retrieval required for climate change predictions. This should be stressed in the introduction and linked with the calibration uncertainty obtained for the Raman lidar. Line 25: Please provide a reference to enhance the importance of such data set in the tropics region. Line 63: "insure"->"ensure" Lines 100-101: Please consider rephrasing this sentence since the overlap function is not only depending on the laser-telescope geometry but also on the entire optical path of the receiving unit according to Kokkalis 2017. 
Panagiotis Kokkalis, Using paraxial approximation to describe the optical setup of a typical EARLINET lidar system, Atmospheric Measurement Techniques, 10, 3103-3115, https://doi.org/10.5194/amt-10-3103-2017, 2017. Line 102-103: "(which ... Lidar1200)". This sentence is redundant. Line 142: The lamp position related to the primary mirror is not affecting the results? Do the authors mean that the white light represents rays coming from infinity (aka rays impinging perpendicular on the primary mirror of the telescope)? Lines 147-148: Are these changes in accordance with changes in the calibration constant of the instrument (and not on the calibration constant of the water vapor retrieval)?
Please clarify more the utilization of the white lamp measurement and what criteria are used to identify changes in the reception part of the system. Line 216: Please indicate here the dead time value of each PMT, and provide a reference on the method used for estimating this value. Lines 274-278: In general, however, it seems that lidar retrieval is strongly underestimated for RH less than 30-40%, something that becomes less obvious for RH values greater than ~60%. Where can be this attributed?
Can it be also due to the overlap height of the apparatus (?), just because the authors mentioned an overlap height earlier ~ 2.2-4 km and here they use values averaged within the first 15 m height.  Line 409: Define this "reasonable uncertainty" here along with the scientific objective behind the chosen value. 

Reviewer 2 Report

Review of Validation of WV profiles of the Raman lidar at Maido Obs (Reunion Island) calibrated with GNSS IWV
by Veremes et al.

Gladly, I have read this interesting and well written paper on highly relevant topic of WV retrieval in upper troposphere and lower stratosphere. I am absolutely intending to give a try to such study with own dataset. The most surprising result is the dependence of the results on the full-moon. The Figures 5 and 6 nicely show that the system has capability to measure well in the given ranges. The paper is very relevant. There is not too many studies on this topic.

I recommend to accept the paper with minor revisions. I do not need to see the revised version. The comments I listed below and many but are not critically influencing the paper and may improve even further this paper.

Abstract
line 11-12 pls change to no significant biases

Intro
line 25 (provide ref.)
line 38 Please comment that the radiosoundings (eg. Foth et al, Stachlewska et al.) are much more often used for calibration than your method or e.g. Dai et al., Sakai et al. or even more exotic ones (eg. Totems & Chazette).
line 39 provide ref. regarding limitations
line 37 and 41- maybe start form new line?
line 42 give more ref. that [25]
line 44 I doubt this book reference can suffice here. There was no discussion on the newer calibration techniques therein. Thus, allocate existing references in better way.
line 46 - name which kind of radiometer you mean
line 52 and 54 repetition regarding 7%
line 63 should read: ensure
line 66 should read: were moved
line 70 indicate wh\t is meant by MLS
line 71 state precisely: simultaneous = all released at the same time.
line 78 should read: needed to be
line 87 should read: is presented

Description of system and database
line 95 remove: When moved to the Maido Obs, and start with The emitted
line 101 confirm: at 1.8km the overlap of the laser beam and the telescope full field of view is being completed.
line 106, 108, 111 optical box = separation unit, then use one of those names throughout the manuscript
line 108 should read: A 2mm diaphragm resulting in 0.5 mrad FOV
line 109 I agree with the sentence but pls indicate that at the same time disadvantage of such narrow FOV is high overlap
line 110 introduce acronym MALICCA-1
line 114 confirm: only PC detection, means issues with signal underestimation in the near range
line 118-119 should read: During a test phase from... to... specific technical studies were conducted.Since Nov 2013, instrument is operated on routine basis.
line 121 should read: based on the use of GNSS
line 124 should read: year-to-year
line 124-125 the no. of msmnts is rather low... time indicates that ou really had single measurement/night. pls comment on whether resampling was done.
line 132 remove consequently (irrelevant here)
line 135 either give the dates for the non-oficial-short-campaing or remove this completely as not relevant to this paper

3. Ancillary msmnts
line 141 what you expect can cause such sudden changes? Stachlewska et al showed long-term stability of the calibration constant and that it is constant over time and changes significantly only if instrumental change was done. do you mean sudden changes or rather typical oscillations around average value?
line 156 should read: in order to record
line 165 should read: The surface pressure
line 169 should read: The GNSS and weather stations have operating rates
line 170-172 Whole sentence is not relevant here - pls remove
line 174 should read: The total uncertainty ranges between
line 191 what is the use of mentioning the STROZ if there is no direct product comparisons exerted? pls comment
line 238 Filtering can prone artefacts for dynamically changing averaging window eg. Stachlewska et al. Did you see any?
Also, is your filter providing exactly the same resolution for the signals at N2 and H2O for each ratio? If not
line 265 really right above ? at 15m? What about the signal underestimation in the lowermost range, this will be due to use of PC only and due

to overlap (~1.8km) and this could be an explanation for the systematically lower values on Fig.1 for dry atmosphere. The fact that this effect

seem to disapear for higher humidity can be explained by general increase of signals for highe humidity (larger/more particles than during the

dry conditions)
line 277 what precisely you mean by the difference in measurement technique.
line 181 if the mail problem would be to reach stratosphere the underestimation of the RH law values derived from lidar would nit depend on range
line 333-334- please check carefully the ratio number and the dates - this does not fit the Fig.3 and Table 1
line 400405 - nice result I did not expected to see such correlation with full-moon. Could you check/add the references if anyone did reported it.
line 422-426 correct point, the automation will help here, e.g. Dai et al.
line 429 well work of the same group is reported here, is there any other references?
line 471 pls consider providing not shown profiles in a supplement

Conclusion
line 512 should read: Since 2013, Raman lidar is operated at the MO (RI).
line 525 pls consider using different name for shooting...
line 526 should read: GNSS is within 5% agreement with the calibration
line 528 remove : can be achieved
line 531 should read: with a total
line 531 should read: measurement over night
line 535 should start with: During MORGANE experiment, the comparison...
line 536-537 confirm: are the same in the given uncertainty range
line 538 well, you do not know if difference is less that WV variability, as you averaged this variability - this is just a result of chosen

averaging!
line 545-562 this should be moved to discussin
line 552 if so, which is actually more limiting - the lidar data availability or the GNSS avaliabilty. pls comment.

Acknlgnt
line 579 will be publicly avaliable? or it is ?
line 586-591 when mentioning personel, either give the full names or the first initials (for consistency)
line 501 replace their with eg. name of station etc.

Appendix A
I appreciate this section but it is rather descriptive - consider providing mathematical formulas
line 606-608 from own experience, this is a lot (15% for atm.profiles), check e.g. Stachlewska&Ritter, from such high uncertainty of atmos

profile is hard to arrive in such low value at 20km, id there mistakt in order of magnitude?
line 611-612 do not get this estimate, how it is done? why only 4% at the ground? if it is so - did you try to cure the data in the overlap range?

Appendix B
line 622 confirm: rolling average is meant here

Figures
Fig.1 did you try to color the valued dependently on season? is there any grouping in the data? In fig.1 caption ple indicate the height of Observatory and the measurement points for lidar and meteo-station.
Fig.3 looking at the figure I have an impression that there are sone trends in this dataeg. from end of Pi till P3 (decreasing), then P3 to P5 increasing p6 decreasing P8-P9 increasing. Please double check the results/interpretation.
Figs.5 and 6 represent very nicely the most important results of this paper; indeed indicate v.dood performance of the Lidar1200!

Equations
General fine but pls in eqs.1 and 2 ramove the '.' for clarity

References
List of the references is complete.
Get aquainted with and consider as additions (not a must!):
https://doi.org/10.5194/amt-12-313-2019
https://doi.org/10.2478/s11600-012-0054-4
https://doi.org/10.5194/amt-11-2735-2018
https://doi.org/10.1016/j.atmosres.2017.05.004
https://doi.org/10.5194/amt-9-1083-2016
https://doi.org/10.5194/acp-10-2813-2010

general:
- remove all 'enters; between the paragrapgs.

- in my version there is g.kg-1 either dot must be in th middle or use g kg-1, same for J pulse-1

- at every place where there is a link to website pls provide the last acess date
